materials science

titanium oxide, nano-size powder, phase transformation, mixed-phase, photocatalyst

**Author for correspondence:**
Takamasa Ishigaki
e-mail: ishigaki@hosei.ac.jp

†Present address: Department of Applied Chemistry, Graduate School of Engineering, Hiroshima University, 1-4-1, Kagamiyama, Higashi-Hiroshima, Japan, 739-8527, Japan.

This article has been edited by the Royal Society of Chemistry, including the commissioning, peer review process and editorial aspects up to the point of acceptance.

# Enhanced visible-light photocatalytic activity of anatase-rutile mixed-phase nano-size powder given by high-temperature heat treatment

Takamasa Ishigaki[1,2,3], Yusuke Nakada[2],
Naoki Tarutani[1,3,†], Tetsuo Uchikoshi[3,4],
Yoshihiro Tsujimoto[4], Masaaki Isobe[4],
Hironori Ogata[1,2,3], Chenning Zhang[4] and Dong Hao[3]

[1]Department of Chemical Science and Technology, and [2]Department of Applied Chemistry, Graduate School of Science and Engineering, Hosei University, 3-5-4 Kajino-cho, Koganei, Tokyo 184-8584, Japan
[3]Research Center for Micro-Nano Technology, Hosei University, 3-11-15 Midori-cho, Koganei, Tokyo 184-0003, Japan
[4]Research Center for Functional Materials, National Institute for Materials Science, 1-2-1 Sengen, Tsukuba, Ibaraki 305-0047, Japan

TI, 0000-0002-7373-5708

Nano-size EVONIK AEROXIDE® P25 titanium dioxide, $TiO_2$, powder was heat-treated at temperatures, 700–900°C, in air. An X-ray diffraction study showed that the P25 powder is composed of approximately 20 and approximately 80 mass% of rutile and anatase phases, respectively. It was also shown that the transformation from anatase to rutile induced by high-temperature heat treatment was almost completed at 750°C, whereas a small amount (less than 3 mass%) of anatase phase was still left even in the powder heat-treated at 900°C. The transformation behaviour was consistent with results obtained by Raman scattering spectroscopy. Raman experiments also indicated that high-temperature heating induced the formation of oxide ion vacancies. Powders were dispersed in methyl orange (MO) aqueous solution, and the bleach rate of MO was measured to evaluate photocatalytic activity under ultraviolet (UV)- and visible-light irradiation. After the heat treatment, the UV-light photocatalytic performance sharply deteriorated. Interestingly, visible-light photocatalytic activity

was enhanced by high-temperature heating and reached the highest performance for an 800°C-heated sample, indicating that the P25 powder obtained high visible-light photocatalytic performance after heat treatment. Even after 900°C heat treatment, the photocatalytic performance was higher than that of as-received powder. Enhancement of photocatalytic activities was discussed in relation to visible light absorption and charge carrier transfer.

# 1. Introduction

Titanium dioxide, $TiO_2$, is a most promising photocatalyst owing to its outstanding properties such as excellent photocatalytic activity, non-toxicity, and long-term thermal, chemical and physical stabilities [1]. Among $TiO_2$ polymorphs, anatase-phase $TiO_2$ is believed to have a superior photocatalytic activity to rutile and brookite. As a photocatalyst, one major disadvantage of $TiO_2$ is that it can only be activated by irradiation with ultraviolet (UV) light, owing to its relatively wide bandgap (approx. 3.2 eV for anatase). As the UV light accounts for only approximately 5% of the solar energy compared to visible light (approx. 45%), any shift in its optical response from UV to the visible spectral range will have a remarkable positive effect on the practical application of the material.

Another problem comes from the transformation from the metastable anatase phase to the thermodynamically stable rutile phase. $TiO_2$ has been widely applied to be coated on tile, ground tile or other surface as functional layers, or to be immobilized on a substrate to construct the structured photocatalyst, in order to meet the requirement for practical applications. Under such situations, high-temperature heat treatment for getting better binding force and preserving high photocatalytic ability after calcination are essential. However, the high-temperature heat treatment gives rise to the transformation. Their photocatalytic activities decrease sharply and are even lost during the phase transformation process from anatase to rutile. Grain growth is also inevitable to occur, resulting in decreased specific surface area and the reduced photocatalytic activities. This has greatly limited the application of $TiO_2$, and it should be crucially important to pursue high-performance $TiO_2$ photocatalysts with both high visible light activity and thermal stability.

Typical commercial nano-size powder, EVONIK(DEGUSSA) AEROXIDE® P25, is composed of rutile and anatase phases (the main crystallographic polymorphisms of $TiO_2$) [2] with a small amount of amorphous phase [3], and anatase and rutile particles in P25 powder are partially interconnected [4,5]. P25 is known to exhibit a high photocatalytic activity and has been often used as a benchmark photocaralyst. According to Huruma et al. [6], mixed-phase titania catalysts show greater photoeffectiveness owing to three factors: (i) the smaller band gap of rutile; (ii) the stabilization of charge separation; and (iii) the small size of the rutile crystallites. The photocatalytic process depends critically on the interface between the $TiO_2$ phases and particle size. The small size of rutile particles in the P25 powder, and the contact with anatase particles that have the comparable size, are crucial to enhancing the catalyst activity.

Porter and his co-investigators reported on calcination of P25 powder [7]. They examined the influence of calcination at temperatures from 600 to 1000°C. High-temperature heat treatment induced the decrease of surface area accompanied with grain growth. Phase transformation from anatase to rutile also took place. Transformation, i.e. increase of rutile content, started above 600°C, and the rutile content was as large as 90 wt% above 750°C. Photocatalytic properties were examined for the degradation of phenol under UV light irradiation. A powder sample heat-treated at 650°C for 3 h revealed highest UV photocatalytic activity, which is slightly higher than that of as-received P25. Further temperature rise led to a steep drop of UV photocatalytic activity. More recently, Wang and his co-investigators reported on calcination of P25 powder at temperatures from 400 to 800°C [8]. Photocatalytic properties under UV irradiation were examined for the decoloration of methyl orange (MO) aqueous solution. Powders heat-treated at temperatures 400–600°C, had higher activity than that of as-received P25 powder. Maximum photocatalytic activity was obtained with a powder heat-treated at 550°C, in which rutile content was bit higher than that of as-received P25 powder. Heat treatment at higher temperatures gave rise to a sharp degradation of UV photocatalytic activity. Inclusion of Si in $TiO_2$ films led the increase of transformation start temperature, and the rutile phase began to form [9]. With the $SiO_2/TiO_2$ composite films, the film heat-treated at 700°C, at which temperature the film still consisted of pure anatase phase, showed the highest decoloration rate of MO aqueous solution under UV-light irradiation. However, visible-light photocatalytic properties were not examined in their works.

Some recent works have reported on heat treatment of titanium oxides [10–14]. As the treatment temperatures in their work were up to 500°C, which is much lower than that in our present work,

coexistence of anatase and rutile was not their main concern. Photocatalytic properties under visible-light irradiation were not concerned, but UV-light photocatalytic characteristics were discussed.

Influence of high-temperature heat treatment on visible-light photocatalytic properties was reported to change the rutile-to-anatase phase content ratio in a $TiO_2$ submicronmetre-size powder synthesized by a sol-gel method [15]. High-temperature heat treatment at 600°C gave a crystallization to generate a pure anatase phase, and the treatment at higher temperatures induced the transformation from anatase to rutile. Photocatalytic activity under visible-light irradiation increased with the increasing heating temperature, and reached a maximum with a powder heat-treated at 750°C of rutile content approximately 60 mass%. To the contrary, the UV-light photocatalytic activity had a maximum for the powder heat-treated at 700°C of rutile content approximately 10 mass%. The variation was explained on the basis of interfacial charge transfer phenomena via rutile-anatase interfaces.

Interconnection between rutile and anatase grains depends on particle size and microstructure. In this work, mixed-phase P25 nano-size powder was heat-treated at temperatures, 700–900°C, at which temperatures the transformation proceeded to increase the rutile phase content, and the photocatalytic activity was examined under UV and visible light irradiation. High-temperature heat treatment gave rise to the sintering of constituent particles, grain growth, phase transformation from anatase to rutile, and formation of oxide ion vacancies. Enhancement of visible-light photocatalytic activities is discussed in relation to phase content and micro-structure.

# 2. Material and methods

Commercial nano-size powder, AEROXIDE® P25 $TiO_2$, purchased from Nippon Aerosil, Ltd., Tokyo, Japan was used for all experiments. The heat-treatment process was performed in a programme control muffle furnace. Samples were heated at given temperature for 3 h in air atmosphere with the ramping rate of 5°C $min^{-1}$ and the cooling rate at 2°C $min^{-1}$, respectively.

Phases were identified by X-ray diffraction (XRD) on a SmartLab X-ray diffractometer (Rigaku Corp., Akishima, Tokyo, Japan) using nickel-filtered Cu K$\alpha$ radiation at 40 kV and 30 mA. Raman spectra (excited at 532 nm) were acquired with a spectroscope (RAMANtouch, Nanophoton Corp., Osaka, Japan). Wavenumber calibration was carried out with a standard Si sample. Particle morphology was observed by a field-emission scanning electron microscope (FE-SEM) (SU8020, Hitachi High-Technologies Corp., Tokyo, Japan). A scanning transmission electron microscope (STEM; JEM-2100F, JEOL Ltd., Akishima, Tokyo, Japan) was employed at an operating voltage of 200 kV to observe microstructures of the samples. For the TEM observation, obtained fine powders were dispersed in ethanol to form a slightly turbid suspension. Ten microlitres of obtained solution was placed on a carbon-coated copper mesh grid and excess solution was removed by swabbing. The grid was dried at room temperature. Diffuse reflection spectra were collected by a UV–visible spectrophotometer equipped with an intergrating sphere (Jasco V-650, Jasco Corp., Hachioji, Tokyo, Japan), in which the baseline was calibrated by using $BaSO_4$. Reflectance data were converted to absorption spectra by Kubelka-Munk transform. X-ray photoelectron spectroscopy (XPS) was performed by an X-ray photoelectron spectrometer (Phi-5600, Physical Electronics, Inc., Minnesota, USA) using monochromatized AlK$\alpha$ at hv = 1486.6 eV. The sample powders were mounted on a polished indium plate and pressed without adhesives. Survey spectra and high-resolution spectra were collected with a step of 0.4 and 0.125 eV, respectively. For each XPS analysis, the sample was exposed to the X-rays for less than 2 h. Spectra were calibrated with C 1 s (284.8 eV) and analysed using PHI MULTIPAK software. Input parameters for the curve resolution procedure included the number of peaks and the peak intensity, peak width at half maximum, and position for each individual peak. Porous characteristics of samples were analysed by $N_2$ sorption measurements (Belsorp-18 II, MicrotracBEL Corp., Osaka, Japan). Prior to $N_2$ sorption measurements, samples were outgassed under vacuum at 200°C for 6 h. The specific surface area was estimated according to the Brunauer–Emmett–Teller method, and the pore size distribution was calculated using the Barrett–Joyner–Halenda method.

Photocatalytic activity was evaluated by bleaching 20 µM of MO (reagent grade, Wako Pure Chemical Industries, Ltd, Osaka, Japan) solution containing 5 mg of powder. To attain the absorption equilibrium of MO, the suspension was stirred ultrasonically for 60 min, and the absorption at the time after 60 min ultrasonic dispersion was set to be that of time zero. Then, UV- and visible-light irradiation was carried out for 120 min at room temperature. Data samplings were made after the light irradiation for 10, 20, 30, 60, 90 and 120 min. The UV light was generated by a UVF-203S Type-A light source (San-Ei Electric Co., Ltd., Osaka, Japan) with a wavelength of 365 nm. The visible light was produced by a UVF–203S Type-C light source (San-Ei Electric Co., Ltd., Osaka, Japan) with a composition of two wavelengths of 405 and

436 nm. After irradiation, the suspension was centrifuged at 12 000 rounds min$^{-1}$ for 30 min by centrifuge (Sigma 2–16, Germany) for separating the powders from the suspension. Herein, the photocatalytic performance was evaluated by using MO rather than methylene blue (MB), owing to the decoloration of MB only from UV irradiation even without any photocatalysts [16]. No decoloration of MO aqueous solution was confirmed after irradiation of UV- and visible-light. The strong absorption peak at the wavelength of 465 nm for the received decolorized MO solution was used to evaluate the photocatalytic performance of the powder, while the absorption intensity at the wavelengths of irradiated lights, 365, 405 and 436 nm, are negligibly weak. Decrease of absorption peak intensity at 465 nm in the absence of dispersed TiO$_2$ particles was confirmed to be less than 1% after 120 min irradiation. The procedure for evaluating the photocatalytic performance is described elsewhere [17].

# 3. Results and discussion

## 3.1. Grain growth and phase transformation

Figure 1 shows the XRD patterns of as-received P25 powder and powders treated at various temperatures of 700–900°C. The P25 powder is synthesized by flame hydrolysis [2]. As-received powder show well-defined diffraction peaks, owing to the high synthesis temperature generated by the oxyhydrogen flame, whose temperature reaches approximately 2700°C. At such high temperature, homogeneous nucleation takes place in a gas phase and the reaction at the surface of existing TiO$_2$ particles follows [18]. The P25 powder is composed of mixed polymorphs, anatase (main phase) and rutile. The metastable anatase is considered as the main phase during a rapid cooling process. According to Skapski, the homogeneous nucleation takes place at much lower temperature than melting temperature under substantially high undercooling [19]. During the synthesis by oxyhydrogen flame, nucleation rate of the anatase should be faster than that of the rutile by comparing their critical nucleation energies, and $\Delta G^*_{\text{anatase}}$, that is, $\Delta G^*_{\text{rutile}} / \Delta G^*_{\text{anatase}} > 1$ [20]. As a result, it is possible to synthesize powders composed of the main metastable phase and minor rutile phase. It was found that, for the P25 powders after the heat treatment at 700–900°C$\Delta G^*_{\text{rutile}}$, the relative intensities of the anatase (101) diffraction peaks decreased, whereas those of the rutile (110) diffraction peaks increased. These opposite trends indicate that a phase transformation from anatase to rutile occurred at high temperature. The transformation was almost completed at 750°C. Relatively low transformation temperature may come from the nano-size particle size. As no obvious orientation can be identified in figure 1, phase composition was evaluated using an equation, $f_R = 1 / (1 + 0.79 \, (I_R / I_A))$, where $f_R$, $I_R$ and $I_A$ denote mass fraction of the rutile phase, $I_R$ and $I_A$, integrated diffraction intensities of rutile (110) and anatase (101), respectively [21]. Rutile content was evaluated to be 17 mass% for the as-received P25 powder, and 68, 98, 98, 95 and 97 mass% for those heated at 700, 750, 800, 850 and 900°C, respectively, although values evaluated for powders heated above 750°C contains relatively large uncertainly and it is just shown that a small amount of anatase phase is still left.

As shown in figure 2, Raman spectra was further used for analysing the phase structure and transformation of TiO$_2$. The solid and dashed lines in figure 2 show their characteristic Raman bands of anatase- and rutile-phases, respectively [22]. The intensity of anatase $E_g$ Raman mode (approx. 145 cm$^{-1}$) decreased with the increase of heating temperature, because the anatase phase was transformed to the rutile phase. Even at the highest heating temperature, 900°C, an anatase peak can be still observed at 145 cm$^{-1}$, illustrating that a small amount of anatase phase existed. Although a rutile peak at 143 cm$^{-1}$ was reported [22], it is thought that the main contribution comes from the anatase 145 cm$^{-1}$ peak. Comparing the peak intensity with that of the main rutile peak at 445 cm$^{-1}$, the peak intensity is too high to think of it as a weak rutile peak. Rutile $E_g$ Raman mode (approx. 445 cm$^{-1}$) was not detected in the as-received sample although 17 mass% of rutile phase was contained. This could be owing to low sensitivity of this Raman mode. High-temperature heating led to the appearance of the rutile peak. With increasing heating temperature, intensities of rutile $E_g$ mode increased, indicating that the phase transformation progressed. Above phase, the transformation result is consistent with the XRD results.

Figure 3 reveals the FE-SEM images of the variation in particle morphologies of P25 commercial powder: (*a*) as-received powder, and those heated at (*b*) 700, (*c*) 750, (*d*) 800, (*e*) 850 and (*f*) 900°C, respectively. As revealed from figure 3*a*, a majority of the as-received nano-sized (approx. 30 nm) particles have a faceted shape. This morphological trend can be ascribed to the super-high processing temperature and super-fast quenching rate of the oxyhydrogen flame, which caused the particles to form directly from gas-solid course.

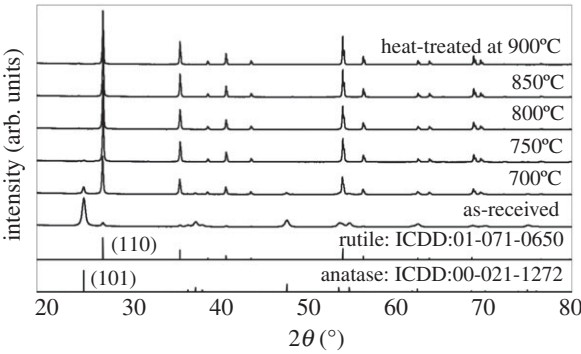

**Figure 1.** XRD patterns of as-received P25 powder and powders heat-treated at various temperatures, 700–900°C.

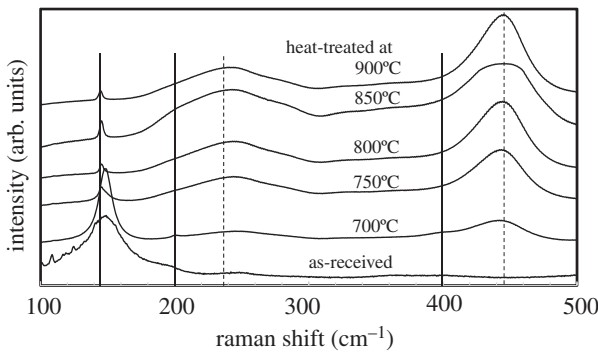

**Figure 2.** Raman spectra of as-received P25 powder and powders subsequently heat-treated at various temperatures, 700–900°C. Reported typical wavenumbers for Raman bands of stoichiometric anatase and rutile are marked with solid and dashed lines, respectively [15].

In figure 4, TEM images of as-received P25 powder are shown. Selected area electron diffractions of figure 4a show the coexistence of anatase and rutile phases. Bright-field image of figure 4b clearly shows the nano-sized particles of faceted shape. Dark-field images taken by the electron diffractions of (101) for anatase (figure 4c) and (110) for rutile (figure 4d) shows that particles are of either anatase or rutile phase, i.e. no anatase/rutile composite particle was observed, and that particles consist of single or a few grains. The observed condition reflects the homogeneous nucleation from a vapour phase to form particles of individual phases, as reported in the vapour phase synthesis of $TiO_2$ nano-size particles via thermal plasma processing, which involves a plasma flame of tremendously high temperature, greater than 10 000 K. Nucleation from a vapour phase and the particle growth take place in the plasma tail region [23]. High-resolution images (figure 4e) taken on the square area in figure 4b shows the stack of single-crystal anatase and rutile particles, i.e. particles are softly agglomerated as shown in the SEM image of figure 4a.

As shown in figure 3 of SEM images, high-temperature heating from 700 to 900°C gave the change of morphology of the soft agglomerated as-received powder, (a), to those of aggregated particles, (b), (c) and (d), and then consolidated bodies, (e) and (f). TEM images of powder heat-treated at 700°C (figure 5) show more clearly the necking between particles. Selected area electron diffractions of figure 5a show the coexistence of anatase and rutile phases. Bright-field image of figure 5b shows increase of particle size, which accords to the SEM image of figure 3b, and the aggregation, i.e. solid-state reaction, between adjacent particles. It is also shown that each primary particle consists of a small number of grains, considering dark-field images of figure 5c,d. High-resolution images (figure 5e) taken on the interconnection of adjacent anatase and rutile grains clearly shows the aggregated condition.

Size distribution of the primary particles was determined from FE-SEM micrographs of figure 3 analysed by using image analyser software. The $d_{50}$ value was taken as an average particle size by counting at least 500 particles. The dependence of average particle size on heating temperature for the P25 $TiO_2$ powder is exhibited in figure 6. A significant increase in particle size occurred from the heat treatment. Particle size of the as-received powder, approximately 30 nm, substantially increased to much larger grain size, approximately 150 nm, after heating at 750°C for 3 h.

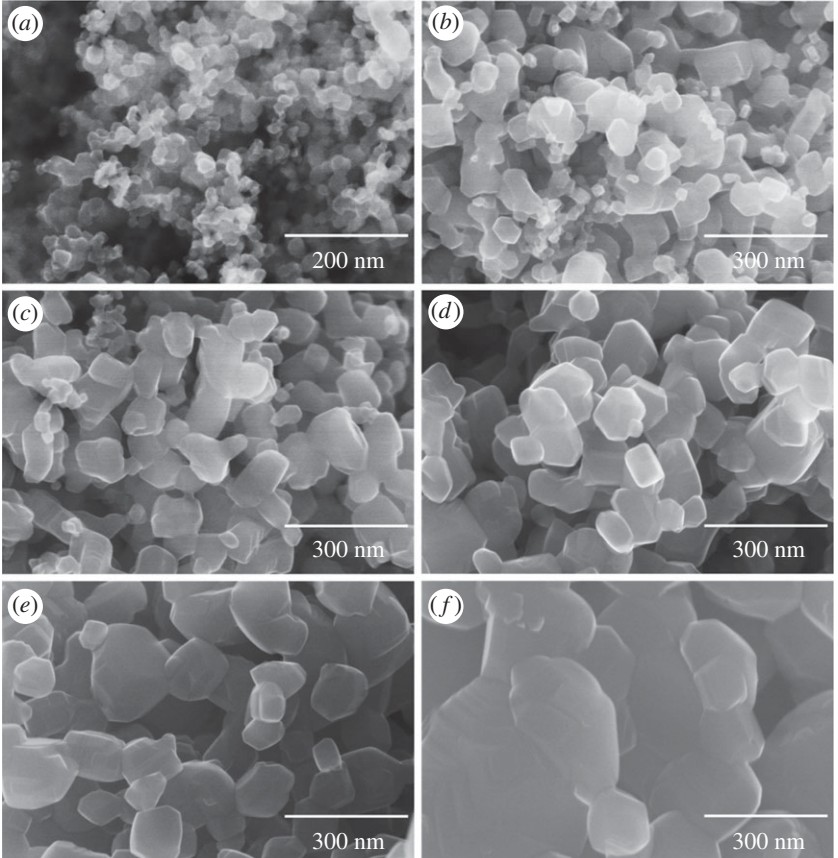

**Figure 3.** Change of particle morphologies of P25 powder; (*a*) as-received powder, and powders heat-treated at (*b*) 700℃, (*c*) 750℃, (*d*) 800℃, (*e*) 850℃ and (*f*) 900℃, respectively.

Change of microstructure, such as the sintering among constituent particles and the subsequent grain growth, led to the decrease of pore volume as well as that of surface area. As is shown in the electronic supplementary material, table S2, the relatively large specific surface of as-received P25 powder, $44.0 \, \mathrm{m^2 \, g^{-1}}$ was steeply decreased by high-temperature heat treatment to 2.7 and $1.7 \, \mathrm{m^2 \, g^{-1}}$ of powders heat-treated at 700 and 800°C, respectively. The sorption isotherm for as-received P25 of electronic supplementary material, figure S2(a) showed the type H3 hysteresis loop [24]. Pore volume also showed a substantial decrease from $0.228 \, \mathrm{cm^3 \, g^{-1}}$ of as-received P25 powder to 0.083 and $0.068 \, \mathrm{cm^3 \, g^{-1}}$ of powders heat-treated at 700 and 800°C, respectively. Evaluated pore volume distribution for the as-received powder shown in electronic supplementary material, figure S2(b) has a broad peak in the range of pore diameter, 50–70 nm, which roughly corresponds to the size of the void in loosely stacked or softly agglomerated P25 particles. Calcination led to the decrease of pose size, approximately 50 nm, although the peak is not so obvious.

## 3.2. Visible-light absorption

Chemical composition and bonding nature were examined by XPS. Survey XPS spectra provided in the electronic supplementary material, figure S3 show powders contain Ti and O, while a trace peak of Cl can be seen only in the spectrum of as-received P25 powder. The XPS peak for C1 s at 284.8 eV is observed owing to the absorbed hydro carbon. Figure 7 shows narrow scan XPS for Ti $2p_{3/2}$ and Ti $2p_{1/2}$ of as-received P25 powder and powders heat-treated at 700 and 800°C. For as-received P25 powder, the featured peaks of 459.3 eV of Ti $2p_{3/2}$ and 465.0 eV of Ti $2p_{1/2}$ are consistent with $Ti^{4+}$ in the $TiO_2$ lattice [25]. Both peaks could be well fitted with each one peak of $Ti^{4+}$. Heat treatment led to the low energy shift of the Ti $2p_{1/2}$ peak. The peak shift with the increase of rutile content was explained by band alignment at the anatase/rutile interface [26]. XPS results agreed with the near intrinsic characteristic of P25 $TiO_2$ powder discussed in the electronic supplementary material, Chapter 1. Chemical analysis showed the concentration of low-valence transition metal in P25 is in the order of 0.01 atm%. Magnetization data of the electronic supplementary material, figure 1S showed the quite low concentration of $Ti^{3+}$.

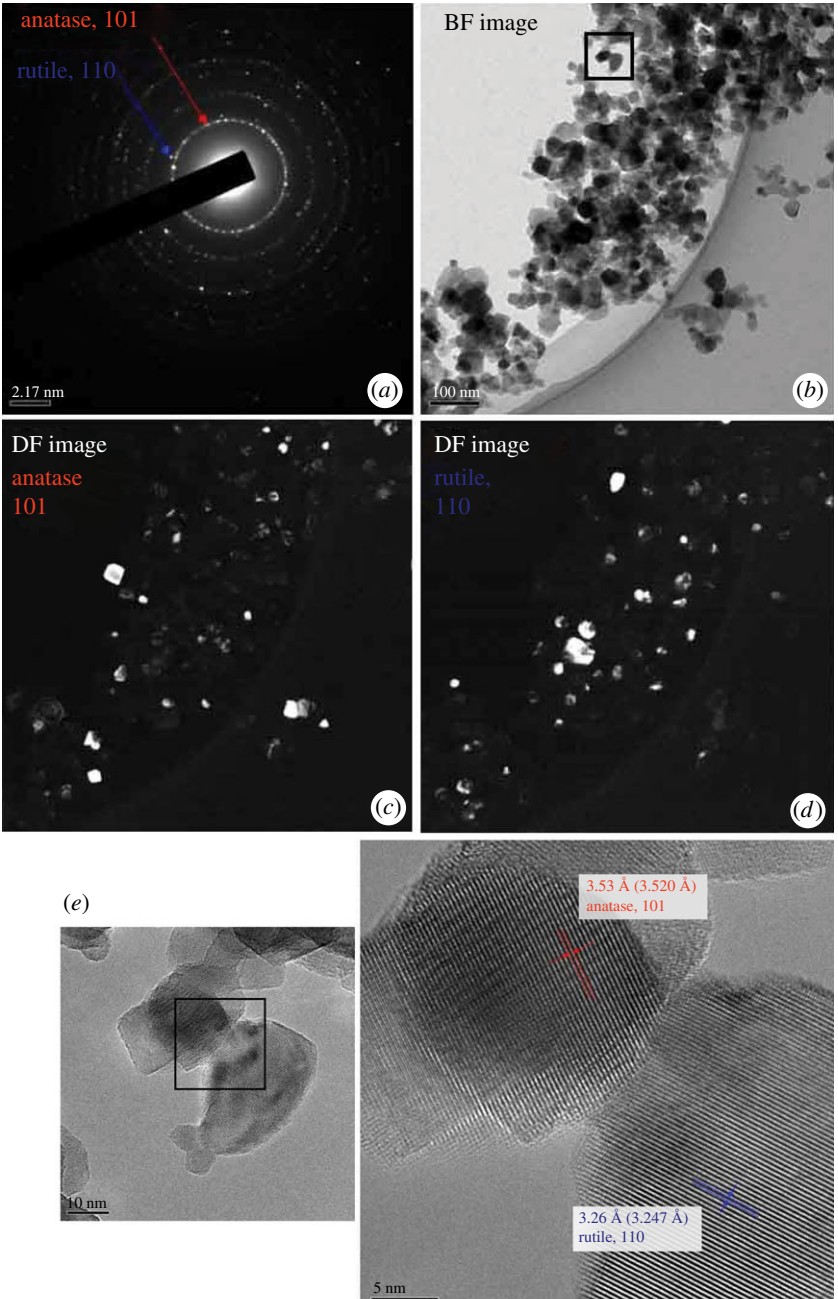

**Figure 4.** TEM images of as-received P25 powder; (*a*) selected area electron diffractions, (*b*) bright-field (BF) image, (*c*,*d*) dark-field (DF) images taken by the electron diffractions of (101) for anatase or (110) for rutile, respectively, (*e*) high-resolution images taken on the square area in (*b*) to show stacked particles.

P25 powder contains small amount of Cl (in the order of $10^{-3}$ mole or less per one mole $TiO_2$), which should substitute oxide ion in the $TiO_2$ lattice [27], as $TiCl_4$ is employed as a precursor for high-temperature flame pyrolysis, which generates highly chemically reactive species, and thus enhances Cl doping. The Cl concentration was semi-quantitatively determined by temperature desorption spectroscopy (TDS). The evaluation procedure for Cl concentration is given in the electronic supplementary material. The TDS spectrum for P25 powder also showed that most of the Cl desorption (greater than 90%) took place up to 600°C, and that the desorption was almost completed at approximately 900°C. Desorbed Cl ions leave vacancies of oxygen sites in the $TiO_2$ lattice. Figure 2 indicates that oxide ion vacancies were formed in $TiO_2$. Raman signal is often very sensitive to the changes of chemical bonding as well as the constituent phase. The increase of oxide ion vacancy concentration gave rise to shifting of the approximately 143 cm$^{-1}$ Raman band ($E_g$) for anatase to the higher wavenumber, and the approximately 447 cm$^{-1}$ Raman band ($E_g$) for rutile to the lower

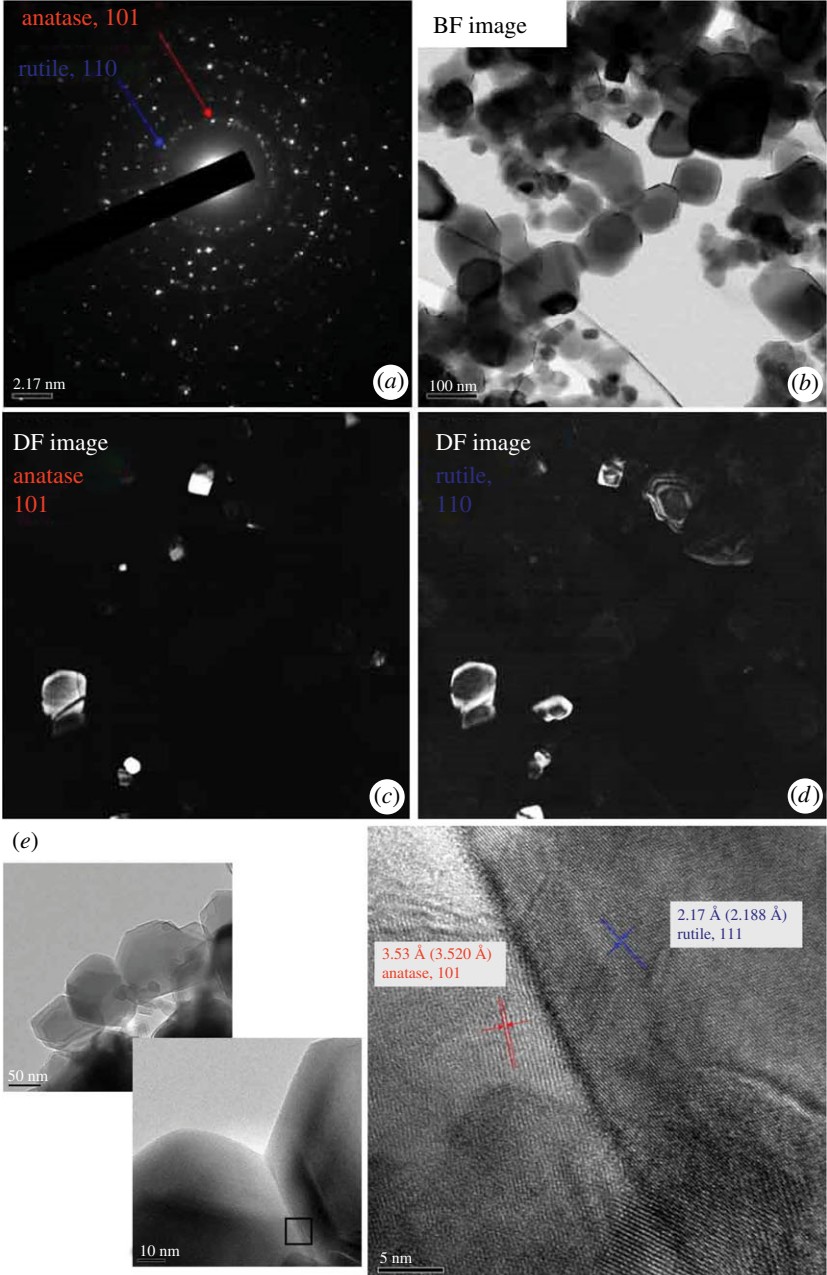

**Figure 5.** TEM images of powder heat-treated at 700°C; (*a*) selected area electron diffractions, (*b*) bright-field (BF) image, (*c*,*d*) dark-field (DF) images taken by the electron diffractions of (101) for anatase or (110) for rutile, respectively, (*e*) images of particle interconnections and high-resolution images taken on the square area.

wavenumber, respectively [28]. In figure 2, the anatase $E_g$ mode (148 cm$^{-1}$) of as-received P25 powder has a slightly higher wavenumber than that of stoichiometric $TiO_2$. The anatase peak of heat-treated powder at 700°C has still higher wavelength number and further heating in air gave the decrease of concentration of oxide ion vacancies, and the Raman shift to that approaching stoichiometric $TiO_2$. To the contrary, the rutile $E_g$ mode (approx. 442 cm$^{-1}$) of as-received P25 powder cannot been seen owing to the relatively small content. The rutile peaks of heat-treated powders at 700 and 800°C have a lower wavelength number than that of stoichiometric $TiO_2$. Further heating gave the increase of oxide ion vacancies, and powder heat-treated at 850°C gives the broad peak of wavenumbers from 438 to 447 cm$^{-1}$. The broad peak was observed with the $TiO_2$ of some amount of oxygen deficiency [29]. Heating at higher temperature led to the decrease of concentration of oxide ion vacancies, and the peak wavenumber of powder heat-treated 900°C is close to that of stoichiometric $TiO_2$. The difference in concentration of oxide ion vacancies in the two polymorphisms is consistent with the fact that the rutile structure has greater tolerance than the anantase structure towards oxide ion vacancies [29,30].

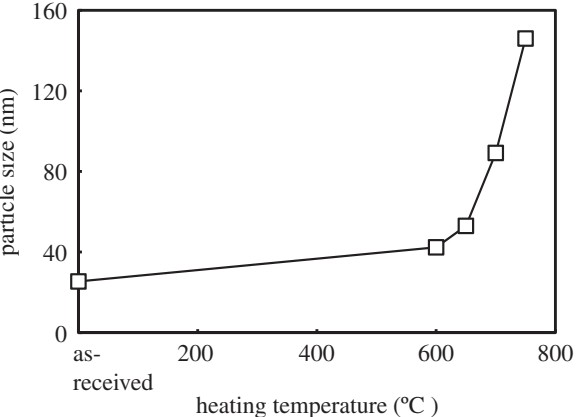

**Figure 6.** Variation in primary particle size of as-received P25 powder and powders heat-treated at various temperatures, 600–750°C evaluated by an image analysis on SEM images. Particle size for heat-treated powders above 800°C could not be determined, as sintering among particles proceeded significantly and primary particles could hardly be recognized.

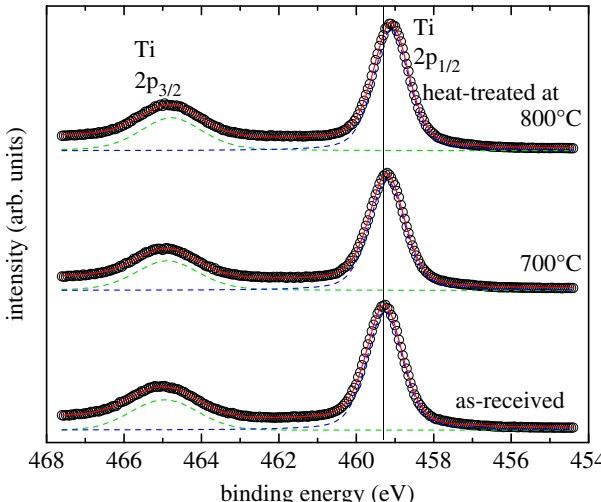

**Figure 7.** Narrow scan XPS for Ti $2p_{3/2}$ and Ti $2p_{1/2}$ of as-received P25 powder and powders heat-treated at 700 and 800°C.

UV–vis absorption spectra converted from diffuse reflection spectra of the P25 powders before and after heat treatment are given in figure 8. As seen from figure 8a, the absorption spectra of heat-treated powders showed a red shift. Especially, when heating the temperature to above 800°C, the onset extended to approximately 420 nm wavelength, making them responsible for visible-light irradiation. Values of band-gap energy ($E_g$) for all powder samples before and after the heat treatment at 700–900°C were estimated. TiO$_2$ has generally been known to be an indirect semiconductor [31]. In an indirect band-gap semiconductor, the minimum of the lowest conduction band is shifted relative to the maximum of the highest valence band (VB), and the lowest-energy interband transition must then be accompanied by phonon excitation [32]. $E_g$ was calculated according to the relationship between the absorption coefficient and incident photon energy for the allowed transition [33,34]:

$$\alpha = \frac{B_i(h\nu - E_g)^2}{h\nu}, \tag{3.1}$$

$$A = \frac{\alpha}{B_i} \tag{3.2}$$

and

$$h\nu = \frac{1240}{\lambda}, \tag{3.3}$$

where $\alpha$ is the absorption coefficient, $h\nu$ is the incident photon energy, $B_i$ is the absorption constant for the indirect transition and $A$ is the absorption proportional to $\alpha$. $E_g$ was obtained by extrapolating the linear parts of the curves in the plots of $(Ah\nu)^{1/2}$ versus $h\nu$, so-called Tauc plots of figure 8b. Evaluated band-gap energies, $E_g$, reduced with increasing the heating temperature; that is, from

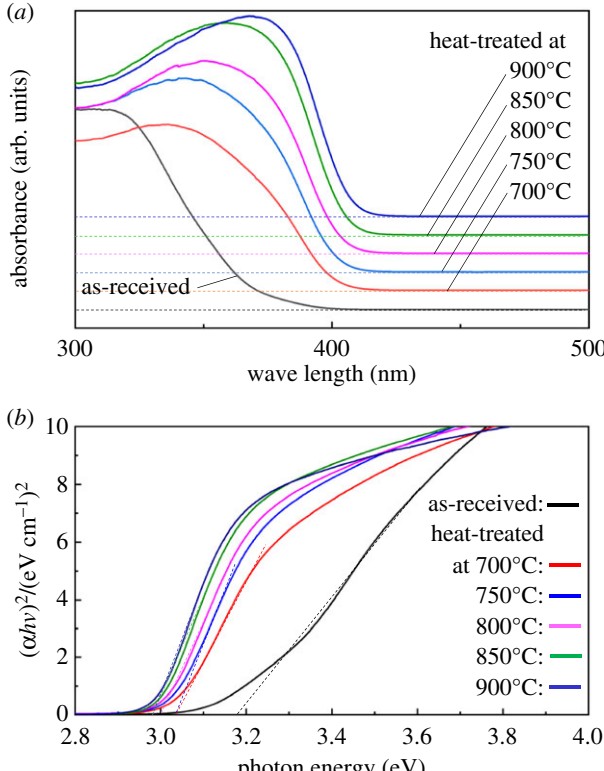

**Figure 8.** (*a*) UV–vis absorption spectra, of as-received P25 powder and powders heat-treated at various temperatures, 700–900°C, converted from diffuse reflectance spectra by Kubelka–Munk transform. (*b*) Tauc plots to evaluate optical band gaps.

3.18 eV for the as-received P25 powder, to 3.02, 3.02, 3.00, 2.98 and 2.98 eV for the heat-treated sample at 700, 750, 800, 850 and 900°C, respectively. This reduction is ascribed to the increase of rutile content, as rutile has a smaller $E_g$ value, approximately 3.0 eV, than those of anatase, and the formation of oxide ion vacancies, as revealed by the Raman spectra in figure 2. The formation of oxide ion vacancies in the $TiO_2$ host lattice has been reported to induce visible-light absorption and therefore improve photocatalytic performance under visible-light irradiation [35–37]. The formed oxygen vacancies can create an impurity level between the bottom of the conduction band (CB) and the top of VB of $TiO_2$ and induce a narrow band gap. It has been proposed that the oxygen-vacancy state is involved in a new photoexcitation process; that is, an electron may be excited to this impurity level from the VB even by the lower energy of visible light and produce the photogenerated charge carriers of $h_{vb}^+$ and $e_{cb}^-$ [38]. Therefore, heat-treated powders should be responsible for the visible-light absorption.

## 3.3. Enhancement of visible-light photocatalytic activity by high-temperature heat treatment

An extended Langmuir–Hinshelwood (L–H) model for a liquid-solid system [39] was applied to the photocatalytic reaction in this experiment. When the initial concentration of the reactant, $C_0$, is very small, the reaction rate is shown by the following equation:

$$\ln\left(\frac{C_0}{C}\right) = k_r t, \tag{3.4}$$

where $C$, $t$ and $k_r$ are the concentration of the reactant, the time, and the apparent first-order reaction-rate constant, respectively. In this work, the concentration of MO used for evaluating photocatalytic performance was 20 µM (a very dilute concentration); thus, $k_r$ was used to indicate the photocatalytic performance of the tested powders. This kinetic model has been successfully applied to various heterogeneous photocatalysis systems [9].

Figure 9 shows the dependence of the MO-bleaching rate of the P25 powders before and after high-temperature treatment under UV- and visible light-irradiation on treatment temperature. As-received P25 powder shows the relatively high UV-light photocatalytic activity. The anatase content, which is commonly viewed as more active than rutile-$TiO_2$ in terms of photocatalytic performance [40], was 83

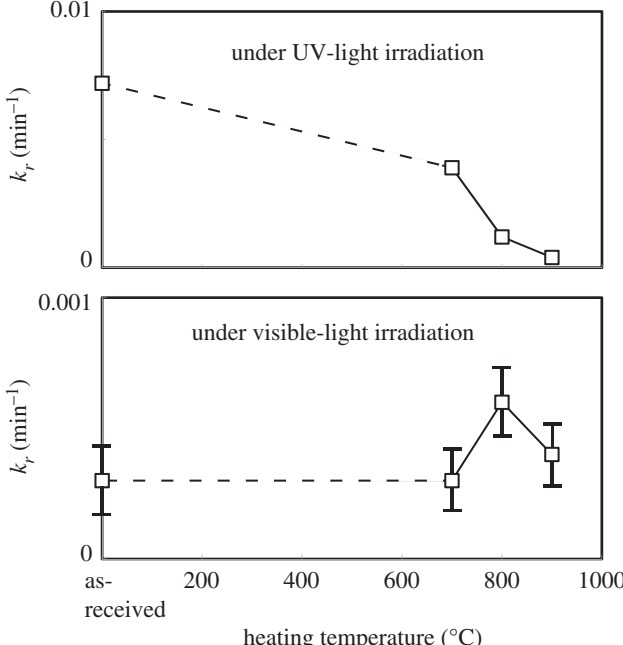

**Figure 9.** Variation in MO-bleaching performance of as-received P25 powder and powders heat-treated at various temperatures, 700–900°C under UV- and visible-light irradiation for 120 min with heat-treatment temperature. Apparent first-order reaction-rate constant, $k_r$, was determined according to equation (3.4). Error bars show standard deviations of rate constant. Values of standard deviations under UV-light irradiation was smaller than corresponding values to the sides of the squares in the figure, and error bars were not drawn.

mass% for P25. Moreover, the interconnection between these two phases allows for rapid electron transfer from rutile to anatase [6]. After the heat treatment, the photocatalytic performance sharply decreased. The particle size of the P25 TiO$_2$ significantly grew after treatment at 800°C. Specific surface area (2.7 m$^2$ g$^{-1}$ after 800°C-treatment) of the P25 powder seriously shrank and, therefore, the number of surface sites, which facilitated access, adsorption and decolorization of MO, dramatically reduced. Moreover, extremely low anatase content owing to the anatase-to-rutile transformation after heat treatment should be considered to explain the reason for steep deterioration of photocatalytic performance of the P25 powder. After the heat treatment, anatase content in the phase composition of the P25 powder was dramatically decreased from 83 mass% (for the as-synthesized powder) to 32 mass% (for the 700°C-treated one) and less than 3 mass% (for 800°C-treated one).

Interestingly, the visible-light photocatalytic performances of the P25 TiO$_2$ powders was enhanced by the thermal treatment. Before the heat treatment, the P25 sample shows very low photocatalytic performance under visible light irradiation. It was found that the breaching rate constant, $k'$, was increased by high-temperature annealing and it increased to a maximum value after the 800°C heat treatment, at which temperature the surface area was significantly small compared to that of as-synthesized P25 powder, indicating that the P25 powder possessed high photocatalytic performance after heat treatment. The photocatalytic performance under visible light irradiation was enhanced with increasing heat-treatment temperature; however, when elevating temperature to 900°C, the performance deteriorated. Even after 900°C heat treatment, a very small amount of anatase is left in the powder, and the photocatalytic performance is higher than the as-received powder.

It has been assumed that the photocatalytic activity of TiO$_2$ photocatalyst should be lowered and even lost after high-temperature calcination owing to the decrease of specific surface areas caused by grain growth as well as phase transformation from anatase to rutile, even though a report showed the enhanced visible-light photocatalytic activity by high-temperature heating [15]. In the present work, the P25 powder showed even higher visible light photocatalytic activity after high-temperature treatment than that of as-received powder. The mechanism can be explained from two aspects, namely, band gap narrowing and electron transfer.

Raman data in figure 2 indicated the formation of oxide ion vacancies. Nakamura *et al*. found that the oxide ion vacancy induced impurity level would act as a trapping site as well as band gap narrowing, to enhance the photogenerated charge transfer [38].

The photocatalytic activity of the anatase-rutile mixed $TiO_2$ under UV light irradiation has been reported to be higher than the single-phase $TiO_2$, because, in regards to the charge carrier migration behaviour in the mixed $TiO_2$, photogenerated electrons and holes are more effectively separated [41]. The rutile/anatase interface is the active site for charge separation as one of the dominant factors in the photocatalytic activity by effectively transferring the photoexcited electrons from the conduction band of anatase to that of rutile [42,43].

More recently, another enhancement mechanism was reported for the anatase-rutile mixed $TiO_2$, in which the electron transfer from rutile to anatase could give higher photoexcited electron-hole separation efficiency [6,44].

Thus, rutile acts as an antenna to extend the photoactivity into visible wavelengths and the structural arrangement of the similarly sized $TiO_2$ crystallites creates catalytic 'hot spots' at the rutile-anatase interface. The rutile phase, which has a relative narrow band gap, will extend the absorbance threshold for harvesting more visible light. After absorbing the light, the photogenerated electrons will jump from the VB of rutile to the CB of rutile. Then the photogenerated electrons can be rapidly transferred to the anatase trapping sites in the crystalline lattice owing to close interjunction between the rutile and anatase phase, and the photogenerated holes can still be located on the rutile phase. This process would effectively promote the charge separation efficiency so as to obtain a higher photocatalytic performance. Therefore, the whole photocatalytic activity was enhanced in the coexistence of anatase and rutile. Also, oxygen vacancies at the junction effectively reduce the band gap as well decrease the carrier recombination to enhance the photocatalytic activity [45]. Thus, mixed phases of anatase and rutile given by high-temperature exhibited the higher photocatalytic activities. The band alignment between rutile and anatase phases was demonstrated for the two-phase interface by the combination of material simulation techniques and XPS experiments [26]. The work showed that the conduction band of rutile lies higher than that of anatase, and that electron transfer from rutile to anatase across the interface should be favourable.

Optimum rutile/anatase ratio in mixed-phase $TiO_2$ for the photocatalytic performance cannot be determined unambiguously. In this work, photocatalytic performance was enhanced to reach an optimum heating temperature, approximately 800°C, and deteriorated by higher-temperature heat treatment. The particle size of the P25 powder significantly increased to several hundreds of nanometres, and the particles showed signs of sintering (figure 4e,f). The surface sites for absorption and reaction of MO molecules was significantly deceased. The rutile content in the powder heated at 800°C was greater than 95 mass%, which is much higher than reported values [15,46]. Optimum rutile content was reported to be approximately 70 and 40 mass%, for submicrometre-size sphere particles and porous bodies composed of submicrometre-size grains, respectively [15,46]. In both samples, crystalline size is 30–50 nm, which is comparable to that in P25 powder. Interconnection between rutile and anatase grains depends on particle size and microstructure as well as crystalline size.

# 4. Conclusion

To pursue the challenge of developing thermal stable $TiO_2$ with high visible light response photocatalytic activity, the thermal stability and visible light photocatalytic activity were investigated on typical nano-size $TiO_2$ powder, P25. As-received P25 $TiO_2$ powder had a mixed composition of polymorphous of anatase (main phase) and rutile. Heat treatment created the formation of oxygen vacancies to generate an impurity level between the conduction and VB in the $TiO_2$ lattice. Also, transformation took place to generate the rutile phase with a smaller band gap than the anatase phase, thereby, narrowing the $TiO_2$ band gap was responsible for visible light absorption. This mixture provided active sites at the rutile/anatase interface for $e^−$–$h^+$ charge separation also thus increasing high visible-light photocatalytic performance. UV-light photocatalytic activity steeply retrograded after high-temperature heat treatment. To the contrary, visible light photocatalytic activity was enhanced, and showed the maximum photocatalytic activity after heat-treated at 800°C.

This study should provide a supporting idea on practical applications of $TiO_2$ into areas, such as self-cleaning coating, environmental purification that need high-temperature heat treatment. It was suggested that elevated transformation temperature and suppression of grain growth should be key factors for the high-temperature stability of $TiO_2$ photocatalysts. We have reported the synthesis of Nb-doped $TiO_2$ nano-size powders via liquid precursor processing in radio-frequency thermal plasma [47]. We demonstrated the extremely high concentration, i.e. much higher than equilibrium doping concentration of Nb into $TiO_2$ that was attained. The Nb doping suppressed the aggregation of

nanoparticles and the phase transformation of anatase to rutile phases both induced by the heat treatment. Influence of high-temperature heat-treatment on visible light photocatalytic activity will be presented for the nano-size Nb-doped $TiO_2$ powders in a further paper.

Data accessibility. Supporting data have been uploaded as part of the electronic supplementary material.

Authors' contributions. T.I. coordinated the study, participated in the design of the study and data analysis, and drafted the manuscript; Y.N. carried out the laboratory work and participated in data analysis; N.T. carried out the laboratory work and participated in data analysis; T.U. and Y.N. participated in the design of the study; M.I. carried out the laboratory work; H.O. participated in the design of the study; C.Z. and D.H. carried out the laboratory work. All authors gave final approval for publication.

Competing interests. We have no competing interests.

Funding. T.I., H.O. and D.H. were supported by a programme, the Strategic Research Foundation at Private Universities, grant no. S1311023, from Ministry of Education, Culture, Sports, Science and Technology (MEXT), Japan. Y.N., N.T., T.U., Y.T., M.I. and C.Z. had no funding support.

Acknowledgements. We thank Mr K. Yamada with NIMS, and Ms A. Okubo and Mr R. Kato with Hosei University for performing chemical composition analysis, XPS and BET measurement, respectively.

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
