## [Reviewer comments · Royal Society Open Science]

Review History

RSOS-181654.R0 (Original submission)

Review form: Reviewer 1

Is the manuscript scientifically sound in its present form?

No

Are the interpretations and conclusions justified by the results?

Yes

Is the language acceptable?

Yes

Is it clear how to access all supporting data?

Yes

Do you have any ethical concerns with this paper?

No

Have you any concerns about statistical analyses in this paper?

No

Recommendation?

Major revision is needed (please make suggestions in comments)

Comments to the Author(s)

Herein, nano-size EVONIK AEROXIDE® P25 TiO₂ powder was heat-treated at temperatures, 700-900°C, in air. X-ray diffraction study showed that the P25 powder is composed of ~20 and ~80 mass% of rutile and anatase phases, respectively. It was also shown that the transformation from anatase to rutile induced by high-temperature heat-treatment was almost completed at 750°C, whereas small amount (<3 mass %) of anatase phase was still left even in the powder heat-treated at 900°C. After the heat treatment, the UV-light photocatalytic performance sharply deteriorated. Interestingly, visible light photocatalytic activity was enhanced by high-temperature heating and reached the highest performance for 800°C-heated sample, indicating that the P25 powder obtained high visible-light photocatalytic performance after heat treatment. The work is interesting. I recommend it accepted for publication after some revising. The main concerns are as follows:

1 To determine the chemical composition, elemental chemical status and microstructure of the prepared samples, more characterization including BET, pore size distribution, TEM, XPS etc. can be added.

2 Some new and related references were overlooked.

CHEMCATCHEM Volume: 10 Issue: 16 Pages: 3469-3480 Published: AUG 21 2018

APPLIED SURFACE SCIENCE Volume: 462 Pages: 480-488 Published: DEC 31 2018

APPLIED SURFACE SCIENCE Volume: 430 Pages: 407-414 Published: FEB 1 2018

MATERIALS RESEARCH EXPRESS Volume: 6 Issue: 2 Article Number: 025039
Published: FEB 2019

INTERNATIONAL JOURNAL OF HYDROGEN ENERGY Volume: 39 Issue: 28 Pages:
15394-15402 Published: SEP 23 2014

Review form: Reviewer 2

Is the manuscript scientifically sound in its present form?

No

Are the interpretations and conclusions justified by the results?

No

Is the language acceptable?

Yes

Is it clear how to access all supporting data?

Yes

Do you have any ethical concerns with this paper?

No

Have you any concerns about statistical analyses in this paper?

Yes

Recommendation?

Reject

Comments to the Author(s)

The authors present the results of experiments in which commercial Evonik Aeroxide P25 (a.k.a. Degussa P25) TiO₂ powder was calcined in air at various temperatures in the range 600-900C and the photocatalytic activity was tested. The apparent novelty of this manuscript is that the photocatalytic activity has been tested using both a UV lamp (365 nm) and a visible (405 and 436 nm), and so the authors were able to discern differences in the UV and visible photocatalytic activity of the various powders. The authors concluded that the powders that were calcined above 700C had enhanced visible photocatalytic activity compared to the others. They also showed that the UV photocatalytic activity decreased with calcining temperature.

In my opinion, this paper would only be of moderate interest to the community. I am not aware of a similar study that investigated the visible activity of the TiO₂ after calcining, but I have not made an exhaustive search. However, the effect of calcination on P25 has been studied extensively, so many of the conclusions of this paper are not surprising: It is well known that anatase converts to rutile with calcining and that the bandgap decreases because rutile becomes the dominant phase and because sintering increases the particle size, which also causes a red-shift in the bandgap. Because the bandgap decreases, one would expect enhanced visible photocatalysis compared to anatase.

Porter et al. were probably the first to report on calcination of P25 (J Mater. Sci. 1999, 34, 1523). Their paper has been cited 157 times in the literature, but not by the authors. A more recent example is Wang et al., International J Photoenergy Volume 2012, Article ID 265760, doi:10.1155/2012/265760. Wang's work is very close to the present manuscript, but was not cited. Wang did not use a visible lamp, but found similar patterns to those reported by the current authors, and their report should be cited and discussed. There may be some differences in the specifics; however, it is well known that the calcination program (heating ramp, dwell time) greatly affect the final product, so that is not surprising. The patterns are very similar. In my opinion, the authors have not delved deeply into the literature for their introduction and discussion sections. This should be done before publishing, should the decision be in the authors' favour.

I would also expect the following revisions (or a well-reasoned answer to the criticisms):

(1) On page 2, line 58 the authors state that the absorption of methyl orange is negligible at 436, 405 and 365 nm, compared to 465 nm. I inspected a uv-visible spectrum of MO (from the web) and found that the absorption values were about 84%, 62% and 23% of the value at 465 nm for 436, 405 and 365 nm, respectively. Since the authors claim that there is no decoloration of MO due to the light in the absence of TiO₂, they should provide some data to support that.

(2) The authors claim that the small difference in wavenumber between their rutile peak and that reported in reference 14 indicates the presence of oxygen vacancies. While I believe that oxygen vacancies are possible, the Raman evidence is not strong, and the authors need to use phrasing that is consistent with their data; e.g., "figure 2 suggests that" rather than the more positive position they have taken. If they wish to really nail down the presence or absence of oxygen vacancies they should use a more direct method, such as XPS. I have a number of specific criticisms of the Raman section:

- a) The authors state that their 'as received' sample has a small peak at 444 cm⁻¹. That is not correct; in fact, there is a trough at that wavelength. I stretched the vertical axis of their figure 2 to determine that.
- b) The dashed line at 197 cm⁻¹ should be solid because it belongs to anatase.
- c) There is a solid line at 244 cm⁻¹ that does not correspond to any line in ref. 14. Perhaps it should move to 235 cm⁻¹ and should be dashed to correspond with the rutile combination band.
- d) The dashed line at 448 cm⁻¹ should be moved to 447 cm⁻¹.
- e) The authors make much of the difference between the literature value of 447 cm⁻¹ for the Eg band of rutile and their observed values, 443 cm⁻¹ (700C) to 446 cm⁻¹ (900C); however, they do not say anything about the very obvious broadening at 850C or the difference between their observed anatase peaks (148 cm⁻¹ 'as received' to 144 cm⁻¹ after calcining at 900C) and the value from ref. 14 of 144 cm⁻¹, except to say that it was "not as obvious" as the rutile shift. The two differences are identical, and in opposite directions. This makes their interpretation less tenable in my opinion.
- g) The authors do not mention that rutile has a weak peak at 143 cm⁻¹, which would overlap the anatase peak at 144 cm⁻¹. When anatase is a small proportion of the TiO₂, the rutile peak could be the major contribution to this resonance.
- (3) The authors mention analysing the presence of chlorine by TDS. It seems appropriate to add a figure to show those results.
- (4) The trend in the band gap is not apparent from the diffuse reflectance spectra in figure 5. I would suggest presenting the data as a Kubelka-Monk transformation (absorbance, rather than reflectance) and separating the lines vertically so that they do not overlap and obscure each other, then showing the onsets with intersecting dashed lines (see Wang reference mentioned above).
- (5) Figure 6 should include error bars on the points. The figure caption should include information about the number of replicates performed for each value. The discussion should include a discussion of whether the trends are significant. I also wonder why the values for 750 and 850C are not included in the figure. (In fact, the authors also created samples at 600 and 650 C which were not studied for their photocatalytic activity; other authors have reported enhanced activity.)

Decision letter (RSOS-181654.R0)

14-Jan-2019

Dear Professor Ishigaki:

Manuscript ID: RSOS-181654

Title: "Enhanced visible light photocatalytic activity of anatase-rutile mixed-phase nano-size powder given by high-temperature heat treatment"

Thank you for submitting the above manuscript to Royal Society Open Science. Your paper was sent to reviewers and their comments are included at the bottom of this letter.

In view of the concerns raised by the reviewers, the manuscript has been rejected in its current form. However, a new manuscript may be submitted which takes into consideration these comments. I apologise that this took longer than usual.

Please note that resubmitting your manuscript does not guarantee eventual acceptance, and that your resubmission will be subject to peer review before a decision is made.

Your resubmitted manuscript should be submitted by 14-Jul-2019. If you are unable to submit by this date please contact the Editorial Office.

On behalf of the Subject Editor Professor Anthony Stace and the Associate Editor Professor Eva Hevia

REVIEWER(S) REPORTS:
Associate Editor Comments to Author ():
RSC Associate Editor:
Comments to the Author:
(There are no comments.)

RSC Subject Editor:
Comments to the Author:
(There are no comments.)

Reviewers' Comments to Author:
Reviewer: 1

Comments to the Author(s)
Herein, nano-size EVONIK AEROXIDE® P25 TiO₂ powder was heat-treated at temperatures, 700-900°C, in air. X-ray diffraction study showed that the P25 powder is composed of ~20 and ~80 mass% of rutile and anatase phases, respectively. It was also shown that the transformation

from anatase to rutile induced by high-temperature heat-treatment was almost completed at 750°C, whereas small amount (<3 mass %) of anatase phase was still left even in the powder heat-treated at 900°C. After the heat treatment, the UV-light photocatalytic performance sharply deteriorated. Interestingly, visible light photocatalytic activity was enhanced by high-temperature heating and reached the highest performance for 800°C-heated sample, indicating that the P25 powder obtained high visible-light photocatalytic performance after heat treatment. The work is interesting. I recommend it accepted for publication after some revising. The main concerns are as follows:

1 To determine the chemical composition, elemental chemical status and microstructure of the prepared samples, more characterization including BET, pore size distribution, TEM, XPS etc. can be added.

2 Some new and related references were overlooked.

CHEMCATCHEM Volume: 10 Issue: 16 Pages: 3469-3480 Published: AUG 21 2018

APPLIED SURFACE SCIENCE Volume: 462 Pages: 480-488 Published: DEC 31 2018

APPLIED SURFACE SCIENCE Volume: 430 Pages: 407-414 Published: FEB 1 2018

MATERIALS RESEARCH EXPRESS Volume: 6 Issue: 2 Article Number: 025039

Published: FEB 2019

INTERNATIONAL JOURNAL OF HYDROGEN ENERGY Volume: 39 Issue: 28 Pages: 15394-15402 Published: SEP 23 2014

Reviewer: 2

Comments to the Author(s)

The authors present the results of experiments in which commercial Evonik Aeroxide P25 (a.k.a. Degussa P25) TiO₂ powder was calcined in air at various temperatures in the range 600-900C and the photocatalytic activity was tested. The apparent novelty of this manuscript is that the photocatalytic activity has been tested using both a UV lamp (365 nm) and a visible (405 and 436 nm), and so the authors were able to discern differences in the UV and visible photocatalytic activity of the various powders. The authors concluded that the powders that were calcined above 700C had enhanced visible photocatalytic activity compared to the others. They also showed that the UV photocatalytic activity decreased with calcining temperature.

In my opinion, this paper would only be of moderate interest to the community. I am not aware of a similar study that investigated the visible activity of the TiO₂ after calcining, but I have not made an exhaustive search. However, the effect of calcination on P25 has been studied extensively, so many of the conclusions of this paper are not surprising: It is well known that anatase converts to rutile with calcining and that the bandgap decreases because rutile becomes the dominant phase and because sintering increases the particle size, which also causes a red-shift in the bandgap. Because the bandgap decreases, one would expect enhanced visible photocatalysis compared to anatase.

Porter et al. were probably the first to report on calcination of P25 (J Mater. Sci. 1999, 34, 1523). Their paper has been cited 157 times in the literature, but not by the authors. A more recent example is Wang et al., International J Photoenergy Volume 2012, Article ID 265760, doi:10.1155/2012/265760. Wang's work is very close to the present manuscript, but was not cited. Wang did not use a visible lamp, but found similar patterns to those reported by the current authors, and their report should be cited and discussed. There may be some differences in the specifics; however, it is well known that the calcination program (heating ramp, dwell time) greatly affect the final product, so that is not surprising. The patterns are very similar. In my opinion, the authors have not delved deeply into the literature for their introduction and discussion sections. This should be done before publishing, should the decision be in the authors' favour.

I would also expect the following revisions (or a well-reasoned answer to the criticisms):

(1) On page 2, line 58 the authors state that the absorption of methyl orange is negligible at 436, 405 and 365 nm, compared to 465 nm. I inspected a uv-visible spectrum of MO (from the web) and found that the absorption values were about 84%, 62% and 23% of the value at 465 nm for 436, 405 and 365 nm, respectively. Since the authors claim that there is no decoloration of MO due to the light in the absence of TiO₂, they should provide some data to support that.

(2) The authors claim that the small difference in wavenumber between their rutile peak and that reported in reference 14 indicates the presence of oxygen vacancies. While I believe that oxygen vacancies are possible, the Raman evidence is not strong, and the authors need to use phrasing that is consistent with their data; e.g., "figure 2 suggests that" rather than the more positive position they have taken. If they wish to really nail down the presence or absence of oxygen vacancies they should use a more direct method, such as XPS. I have a number of specific criticisms of the Raman section:

a) The authors state that their 'as received' sample has a small peak at 444 cm⁻¹. That is not correct; in fact, there is a trough at that wavelength. I stretched the vertical axis of their figure 2 to determine that.

b) The dashed line at 197 cm⁻¹ should be solid because it belongs to anatase.

c) There is a solid line at 244 cm⁻¹ that does not correspond to any line in ref. 14. Perhaps it should move to 235 cm⁻¹ and should be dashed to correspond with the rutile combination band.

d) The dashed line at 448 cm⁻¹ should be moved to 447 cm⁻¹.

e) The authors make much of the difference between the literature value of 447 cm⁻¹ for the Eg band of rutile and their observed values, 443 cm⁻¹ (700C) to 446 cm⁻¹ (900C); however, they do not say anything about the very obvious broadening at 850C or the difference between their observed anatase peaks (148 cm⁻¹ 'as received' to 144 cm⁻¹ after calcining at 900C) and the value from ref. 14 of 144 cm⁻¹, except to say that it was "not as obvious" as the rutile shift. The two differences are identical, and in opposite directions. This makes their interpretation less tenable in my opinion.

g) The authors do not mention that rutile has a weak peak at 143 cm⁻¹, which would overlap the anatase peak at 144 cm⁻¹. When anatase is a small proportion of the TiO₂, the rutile peak could be the major contribution to this resonance.

(3) The authors mention analysing the presence of chlorine by TDS. It seems appropriate to add a figure to show those results.

(4) The trend in the band gap is not apparent from the diffuse reflectance spectra in figure 5. I would suggest presenting the data as a Kubelka-Monk transformation (absorbance, rather than reflectance) and separating the lines vertically so that they do not overlap and obscure each other, then showing the onsets with intersecting dashed lines (see Wang reference mentioned above).

(5) Figure 6 should include error bars on the points. The figure caption should include information about the number of replicates performed for each value. The discussion should include a discussion of whether the trends are significant. I also wonder why the values for 750 and 850C are not included in the figure. (In fact, the authors also created samples at 600 and 650

C which were not studied for their photocatalytic activity; other authors have reported enhanced activity.)

Author's Response to Decision Letter for (RSOS-181654.R0)

See Appendix A.

RSOS-191539.R0

Review form: Reviewer 1

Is the manuscript scientifically sound in its present form?

Yes

Are the interpretations and conclusions justified by the results?

Yes

Is the language acceptable?

Yes

Do you have any ethical concerns with this paper?

No

Have you any concerns about statistical analyses in this paper?

No

Recommendation?

Accept as is

Comments to the Author(s)

Publish as is.

Review form: Reviewer 2

Is the manuscript scientifically sound in its present form?

Yes

Are the interpretations and conclusions justified by the results?

Yes

Is the language acceptable?

Yes

Do you have any ethical concerns with this paper?

No

Have you any concerns about statistical analyses in this paper?

No

Recommendation?

Accept as is

Comments to the Author(s)

The authors have answered my comments satisfactorily and have greatly improved the manuscript with additional discussion and references, and have improved the figures and data analysis. It is unfortunate that they are unable at this time to test catalysts prepared at other temperatures; however, the manuscript in its present form is publishable.

Decision letter (RSOS-191539.R0)

28-Oct-2019

Dear Professor Ishigaki:

Title: Enhanced visible light photocatalytic activity of anatase-rutile mixed-phase nano-size powder given by high-temperature heat treatment
Manuscript ID: RSOS-191539

It is a pleasure to accept your manuscript in its current form for publication in Royal Society Open Science. The chemistry content of Royal Society Open Science is published in collaboration with the Royal Society of Chemistry.

RSC Associate Editor
Comments to the Author:
(There are no comments.)

Reviewer(s)' Comments to Author:
Reviewer: 1

Comments to the Author(s)
publish as is.

Reviewer: 2

Comments to the Author(s)
The authors have answered my comments satisfactorily and have greatly improved the manuscript with additional discussion and references, and have improved the figures and data analysis. It is unfortunate that they are unable at this time to test catalysts prepared at other temperatures; however, the manuscript in its present form is publishable.

Appendix A

Reply to the Reviewers' comments

Manuscript ID: RSOS-181654

Title: "Enhanced visible light photocatalytic activity of anatase-rutile mixed-phase nano-size powder given by high-temperature heat treatment"

We are much obliged to the referees for their reviewing our original manuscript entitled above as the first revision. We have carefully amended our manuscript following the referee's kind suggestions. Their reviewing of our revised manuscript would also be gratefully appreciated. Revised portions are indicated by *red-colored font in the revised manuscript*. The numbers of the reference and figure in the response letter are those in the revised manuscript, unless they are specified.

We hope that our revisions are satisfactory for publication.

Reviewer: 1

Herein, nano-size EVONIK AEROXIDE P25 TiO₂ powder was heat-treated at temperatures, 700-900°C, in air. X-ray diffraction study showed that the P25 powder is composed of ~20 and ~80 mass% of rutile and anatase phases, respectively. It was also shown that the transformation from anatase to rutile induced by high-temperature heat-treatment was almost completed at 750°C, whereas small amount (<3 mass %) of anatase phase was still left even in the powder heat-treated at 900°C. After the heat treatment, the UV-light photocatalytic performance sharply deteriorated. Interestingly, visible light photocatalytic activity was enhanced by high-temperature heating and reached the highest performance for 800°C-heated sample, indicating that the P25 powder obtained high visible-light photocatalytic performance after heat treatment. The work is interesting. I recommend it accepted for publication after some revising. The main concerns are as follows:

- 1. To determine the chemical composition, elemental chemical status and microstructure of the prepared samples, more characterization including BET, pore size distribution, TEM, XPS etc. can be added.*

Answer: Material characterization was carried out additionally, including BET, pore size distribution, TEM, and XPS.

Results of BET and pore size distribution were shown in the section 3 of Supporting Information, together with Table S2 and Figure S2. Measurement procedure was given in

the chapter, 3. Materials and Method, too, and discussion was given in in the last paragraph of the section, 4.1. Grain Growth and Phase Transformation.

Results of TEM observation was shown in Figures 4 and 5. Observation procedure was given in the chapter, 3. Materials and Method. Discussion was given in the fourth and fifth paragraphs of the section, 4.1. Grain Growth and Phase Transformation.

XPS results were shown in Figure 7 and Figure S3. Measurement procedure was given in the chapter, 3. Materials and Method, and the section 4 of Supporting Information. Discussion was given in the first paragraph of the section, 4.2. Visible Light Absorption.

2. *Some new and related references were overlooked.*

- 1) *CHEMCATCHER* Volume: 10 Issue: 16 Pages: 3469-3480
Published: AUG 21 2018.
- 2) *APPLIED SURFACE SCIENCE* Volume: 462 Pages: 480-488
Published: DEC 31 2018.
- 3) *APPLIED SURFACE SCIENCE* Volume: 430 Pages: 407-414
Published: FEB 1 2018.
- 4) *MATERIALS RESEARCH EXPRESS* Volume: 6 Issue: 2 Article
Number: 025039 Published: FEB 2019.
- 5) *INTERNATIONAL JOURNAL OF HYDROGEN ENERGY* Volume: 39 Issue:
28 Pages: 15394-15402 Published: SEP 23 2014.

Answer: Suggested recent papers were referred as References 9-13, and comments were described in the sixth paragraph of the chapter, 2. Introduction.

These papers reported on heat-treatment. The treatment temperatures in their works were up to 500 °C, which is much lower than those in our present work, 700-900 °C. Also, only photocatalytic properties under UV irradiation was concerned.

Reviewer: 2

The authors present the results of experiments in which commercial Evonik Aeroxide P25 (a.k.a. Degussa P25) TiO₂ powder was calcined in air at various temperatures in the range 600-900C and the photocatalytic activity was tested. The apparent novelty of this manuscript is that the photocatalytic activity has been tested using both a UV lamp (365 nm) and a visible (405 and 436 nm), and so the authors were able to discern differences in the UV and visible photocatalytic activity of the various powders. The authors concluded that the powders that were calcined above 700C had enhanced visible

photocatalytic activity compared to the others. They also showed that the UV photocatalytic activity decreased with calcining temperature.

In my opinion, this paper would only be of moderate interest to the community. I am not aware of a similar study that investigated the visible activity of the TiO₂ after calcining, but I have not made an exhaustive search. However, the effect of calcination on P25 has been studied extensively, so many of the conclusions of this paper are not surprising: It is well known that anatase converts to rutile with calcining and that the bandgap decreases because rutile becomes the dominant phase and because sintering increases the particle size, which also causes a red-shift in the bandgap. Because the bandgap decreases, one would expect enhanced visible photocatalysis compared to anatase.

Answer: The present paper reports the enhancement of visible light photocatalytic properties of nano-size P25 TiO₂ powder induced by heat-treatment at very high temperature, 800 °C. Authors believe that the present paper should report certainly original results. Importance was added in the last paragraph of the chapter, 2. Introduction.

Rutile TiO₂ itself is known to have much less photocatalytic activity than that of anatase TiO₂, although rutile TiO₂ with a small band gap can absorb a visible light. Formation of anatase/rutile interface gives the higher photocatalytic activity than that of anatase TiO₂ itself. In this work, the enhanced visible light photocatalytic was given by high-temperature treatment. The maximum was seen at the temperature, as high as 800 °C.

More materials information was added to show the chemical composition, elemental chemical status and microstructure of heat-treated samples.

1. *Porter et al. were probably the first to report on calcination of P25 (J Mater. Sci. 1999, 34, 1523). Their paper has been cited 157 times in the literature, but not by the authors.*

Answer: The paper by Porter et al. was cited as Reference 7, and description was given in the fourth paragraph of the chapter, 2. Introduction. In their work, photocatalytic properties were examined for the degradation of phenol under UV light irradiation. Calcination temperatures were from 600 to 800 °C. UV light photocatalytic activity of samples heat-treated from 700 to 900 °C in our work showed almost identical degree of deterioration in their work, compared to that of as-received P25.

2. *A more recent example is Wang et al., International J Photoenergy Volume 2012,*

Article ID 265760, doi:10.1155/2012/265760. Wang's work is very close to the present manuscript, but was not cited.

Wang did not use a visible lamp, but found similar patterns to those reported by the current authors, and their report should be cited and discussed. There may be some differences in the specifics; however, it is well known that the calcination program (heating ramp, dwell time) greatly affect the final product, so that is not surprising. The patterns are very similar. In my opinion, the authors have not delved deeply into the literature for their introduction and discussion sections. This should be done before publishing, should the decision be in the authors' favour.

Answer: The paper by Wang et al. was cited as Reference 8, and description was given in the fourth paragraph of the chapter, 2. Introduction

As is mentioned by the reviewer, photocatalytic properties were examined for the degradation of methyl orange MO aqueous solution under UV-light irradiation only. UV-light photocatalytic performance of heat-treated at temperatures, 400-600 °C, showed higher activity than that of as-received powder. In the temperature range, phase composition is not changed, and the increased photocatalytic activity may be related to the improvement of crystallinity or removal of chlorine as well as the progress of interconnection between particles. Powders heat-treated at 700 and 800 °C showed the decrease of photocatalytic activity. Our work also showed the decreased photocatalytic activity, and the degree of deterioration was almost identical to each other.

In the relatively low temperature range, heat-treatment was not carried out in our work, because we had an intention to do at the higher temperature than the temperature, at which phase transformation starts. The statement was given in the last paragraph of the chapter, 2. Introduction.

3. I would also expect the following revisions (or a well-reasoned answer to the criticisms):

(1) On page 2, line 58 the authors state that the absorption of methyl orange is negligible at 436, 405 and 365 nm, compared to 465 nm. I inspected a uv-visible spectrum of MO (from the web) and found that the absorption values were about 84%, 62% and 23% of the value at 465 nm for 436, 405 and 365 nm, respectively. Since the authors claim that there is no decoloration of MO due to the light in the absence of TiO₂, they should provide some data to support that.

Answer: We confirmed that the decoloration of MO solution was negligible in the absence

of TiO₂ particles. Description was given in the last paragraph of the chapter, 3. Materials and Method.

(2) The authors claim that the small difference in wavenumber between their rutile peak and that reported in reference 14 indicates the presence of oxygen vacancies. While I believe that oxygen vacancies are possible, the Raman evidence is not strong, and the authors need to use phrasing that is consistent with their data; e.g., “figure 2 suggests that” rather than the more positive position they have taken. If they wish to really nail down the presence or absence of oxygen vacancies they should use a more direct method, such as XPS.

Answer: XPS measurement was carried out additionally. Results were shown in Figure 7 and Figure S3. The XPS results shown in Figure 7 did not show the existence of Ti³⁺, and showed that TiO₂ of as-received P25 and heat-treated samples consisted of Ti⁴⁺ exclusively.

As it was mentioned in the second paragraph of the section, 4.2. Visible Light Absorption, Raman signal is often much sensitive to chemical bonding. Formation of oxide ion vacancies was discussed in the section, 4.2., by referring References 28 and 29.

(3) I have a number of specific criticisms of the Raman section:

a) The authors state that their ‘as received’ sample has a small peak at 444 cm⁻¹. That is not correct; in fact, there is a trough at that wavelength. I stretched the vertical axis of their figure 2 to determine that.

Answer: The description in the original manuscript was not correct. The rutile peak cannot be seen with the as-received P25 powder. The description was corrected in the second paragraph of the section, 4.1. Grain Growth and Phase Transformation, and the same description was given in the second paragraph of the section, 4.2. Visible Light Absorption.

Figure 2 was revised according to the reviewer’s comments below. Also, arrows in the original figure were removed for clarification.

b) The dashed line at 197 cm⁻¹ should be solid because it belongs to anatase.

Answer: The dashed line at 197 cm⁻¹ was changed to a solid line in Figure 2.

c) *There is a solid line at 244 cm⁻¹ that does not correspond to any line in ref. 14. Perhaps it should move to 235 cm⁻¹ and should be dashed to correspond with the rutile combination band.*

Answer: The solid line at 244 cm⁻¹ was moved to 235 cm⁻¹ and changed to a dashed line in Figure 2.

d) *The dashed line at 448 cm⁻¹ should be moved to 447 cm⁻¹.*

Answer: The dashed line at 448 cm⁻¹ was moved to 447 cm⁻¹ in Figure 2.

e) *The authors make much of the difference between the literature value of 447 cm⁻¹ for the Eg band of rutile and their observed values, 443 cm⁻¹ (700C) to 446 cm⁻¹ (900C); however, they do not say anything about the very obvious broadening at 850C or the difference between their observed anatase peaks (148 cm⁻¹ 'as received' to 144 cm⁻¹ after calcining at 900C) and the value from ref. 14 of 144 cm⁻¹, except to say that it was "not as obvious" as the rutile shift. The two differences are identical, and in opposite directions. This makes their interpretation less tenable in my opinion.*

Answer: Reference 28 reported that anatase Eg peak at 143 cm⁻¹ shifts to higher wave number and rutile Eg peak rutile at 447 cm⁻¹ shifts to lower wave number, when oxide ion vacancies are formed in TiO₂. Description was corrected to explain the shifts of both Raman peaks. Peak broadening observed with a powder treated at 850 °C suggests the formation of some more oxide ion vacancies, as it was reported in Reference 29. Corrected description was given in the second paragraph of the section, 4.2., by referring References 28 and 29.

g) *The authors do not mention that rutile has a weak peak at 143 cm⁻¹, which would overlap the anatase peak at 144 cm⁻¹. When anatase is a small proportion of the TiO₂, the rutile peak could be the major contribution to this resonance.*

Answer: According to Reference 22, the rutile peak at 143 cm⁻¹ is weak. Comparing the peak intensity at 144 cm⁻¹ with that of 445 cm⁻¹, the main contribution comes from anatase 144 cm⁻¹ peak. Comparing the peak intensity with that of main rutile peak at 445 cm⁻¹, the peak intensity is too high to think it as a weak rutile peak. Description concerning the above was added in the second paragraph of the section, 4.1. Grain Growth

and Phase Transformation.

(3) The authors mention analysing the presence of chlorine by TDS. It seems appropriate to add a figure to show those results.

Answer: TDS results were already reported in our previous paper, Reference 27. Evaluation procedure of Cl concentration in powders using thermal desorption spectroscopy (TDS). It is shown that the evaluated value of chlorine concentration from TDS data is a semi-qualitative one.

(4) The trend in the band gap is not apparent from the diffuse reflectance spectra in figure 5. I would suggest presenting the data as a Kubelka-Monk transformation (absorbance, rather than reflectance) and separating the lines vertically so that they do not overlap and obscure each other, then showing the onsets with intersecting dashed lines (see Wang reference mentioned above).

Answer: Figure 5 of original manuscript was revised to Figure 8 of revised manuscript. Figure 8(a) was made according to reviewer's suggestion. Absorbance spectra was converted from diffuse reflectance spectra by Kubelka-Munk transform. Tauc plots to evaluate optical bandgaps was given in Figure 8(b).

(5) Figure 6 should include error bars on the points. The figure caption should include information about the number of replicates performed for each value. The discussion should include a discussion of whether the trends are significant. I also wonder why the values for 750 and 850C are not included in the figure. (In fact, the authors also created samples at 600 and 650 C which were not studied for their photocatalytic activity; other authors have reported enhanced activity.)

Answer: Error bars were added to plots in the revised figure, Figure 9. As for the reproducibility of decoloration rate, the description was added in the third paragraph of the chapter, 3. Materials and Methods. Data samplings were made after the light irradiation for 10, 20, 30, 60, 90 and 120 minutes, and the decoloration rate was determined for each sampling time. For limited conditions, another sampling after 120 minutes irradiation was carried out to confirm reproducibility of measurement in the present work, although this was not described in the manuscript.

As for the heat-treatment temperatures, our original intention was to examine the

photocatalytic activity for powders heat-treated at much higher temperatures than the transformation start temperature. Thus, the photocatalytic measurement was not carried out for the powders heat-treated at 600 and 650°C. The photocatalytic test for powders heat-treated at 750 and 850°C was not done, because it was thought that a certain level of dependency on treated temperature was obtained through the measurement on powders heat-treated at 700, 800 and 900°C. Lines, which connects data points, were partially changed from solid ones to dotted ones in Figure 9.

Authors actually agree with the reviewer's suggestion of doing tests on powders 600 and 650°C, which would be interesting. However, our present condition does not allow us to put the photocatalytic data in the revised manuscript. Mr. Nakada, my student, in charge of photocatalytic test graduated from our university. Our photocatalytic system was modified last autumn, and, therefore, for consistency of results, measurements on a series of samples needs to be carried out from the beginning. Although a new student started his graduation work on a photocatalytic material, we still need to wait for his learning to get reliable data.

Results of the present work showed the enhancement of visible light photocatalytic activity with powders heat-treated at much higher temperature, 700-900°C, than the transformation start temperature. A sentence was added to show more clearly the significance of the present paper in the third paragraph of the chapter, 4.3. Enhancement of Visible Light Photocatalytic Activity by High-Temperature Heat-Treatment. As it is given in the second paragraph of the chapter, 5. Conclusion, this work provided an idea on the fabrication of self-cleaning coating, environmental purification filters that needs high temperature heat treatment.

Other Revised Points

1. References 14, 23-25, and 28 were additionally referred to clarify discussions. Also, References 3 and 4 were added in Supporting Information.
2. In Acknowledgments, Ms. Okubo and Mr. Kato were listed because of their support to the present work.
3. Part of Authors' Contributions was revised throughout.